# Effect of Two Particle Sizes of Nano Zinc Oxide on Growth Performance, Immune Function, Digestive Tract Morphology, and Intestinal Microbiota Composition in Broilers

**DOI:** 10.3390/ani13091454

**Published:** 2023-04-25

**Authors:** Jianyu Qu, Xixi Zuo, Qiurong Xu, Mengyao Li, Lirui Zou, Ran Tao, Xiangyan Liu, Xianglin Wang, Ji Wang, Lixin Wen, Rongfang Li

**Affiliations:** 1Hunan Engineering Research Center of Livestock and Poultry Health Care, Colleges of Veterinary Medicine, Hunan Agricultural University, Changsha 410128, China; 2Animal Nutritional Genome and Germplasm Innovation Research Center, College of Animal Science and Technology, Hunan Agricultural University, Changsha 410128, China; 3Changsha Lvye Biotechnology Co., Ltd., Changsha 410100, China

**Keywords:** broiler chickens, nano-ZnO, growth performance, immune function, gut microbiome

## Abstract

**Simple Summary:**

Nano-ZnO has emerged as a popular study subject for the addition of trace elements in the manufacture of animal feed. However, the current research has not studied how different particle sizes of nano-ZnO affect the broiler. In the current study, we studied the impact of two different nano-ZnO particle sizes in broilers. Finally, it was discovered that ZnO nanoparticles could enhance broiler growth performance, promote the development of immune organs, enhance generalized immunity, have some anti-inflammatory effects, and enrich beneficial intestinal bacteria. The effects of the above improvements were negatively correlated with the nanoparticles’ particle sizes.

**Abstract:**

The effects of dietary supplementation with two particle sizes of nano zinc oxide (ZnO) on growth performance, immune function, intestinal morphology, and the gut microbiome were determined in a 42-day broiler chicken feeding experiment. A total of 75 one-day-old Arbor Acres broilers were randomized and divided into three groups with five replicates of five chicks each, including the conventional ZnO group (NC), the nano-ZnO group with an average particle size of 82 nm (ZNPL), and the nano-ZnO group with an average particle size of 21 nm (ZNPS). Each group was supplemented with 40 mg/kg of ZnO or nano-ZnO. Our results revealed that birds in the ZNPS group had a higher average daily gain and a lower feed-to-gain ratio than those in the NC group. ZNPS significantly increased the thymus index and spleen index, as well as the levels of serum metallothionein (MT), superoxide dismutase (SOD), and lysozyme (LZM). The ZNPS treatments reduced interleukin (IL)-1β and tumor necrosis factor-alpha (TNF-α) levels and increased IL-2 and interferon (IFN)-γ levels compared to that in the NC group. Additionally, compared with the birds in the NC group, those in the nano-ZnO group had a higher villus height to crypt depth ratio of the duodenum, jejunum, and ileum. Bacteroides increased in the ZNPS group at the genus level. Further, unidentified_Lachnospiraceae, Blautia, Lachnoclostridium, unidentified_Erysipelotrichaceae, and Intestinimonas were significantly increased in the ZNPL group. In conclusion, nano-ZnO improved the growth performance, promoted the development of immune organs, increased nonspecific immunity, improved the villus height to crypt depth ratio of the small intestine, and enriched the abundance of beneficial bacteria. Notably, the smaller particle size (21 nm) of nano-ZnO exhibited a more potent effect.

## 1. Introduction

The second-most prevalent trace element in the body, zinc (Zn), must be ingested from external sources [1]. Zn, a mineral, is important for birds’ nutrition metabolism, appetite regulation, and cleaning up reactive oxygen species [2]. The average daily weight gain, feed conversion rate, and feed intake of broilers can all be considerably increased by adding the appropriate amount of Zn to their feed [3]. In addition, Zn has a significant impact on the immunological system. Zn deficiency can cause immune system atrophy, a severe decrease in immune cell generation [4], and cell-mediated immune dysfunction in animals [5]. Further, clinical studies have suggested that a lack of Zn can also cause dysregulation of the microbiota in chickens [6]. Recently, the intestinal microbiota has received much research interest, and an increasing number of studies have elucidated its crucial regulatory effect on health. Notably, the intestinal microbiota affects various organ systems, including the nervous, immune, digestive, and reproductive systems [7,8,9]. Thus, it is sometimes referred to as the “second brain”.

Zn is added to livestock feed in organic (such as Zn lactate, amino acid Zn, and chelated Zn) and inorganic (such as Zn sulfate and oxide) forms. According to a study, zinc threonine chelate has a higher bioavailability of zinc than zinc sulfate [10]. Owing to the high cost of organic Zn, traditional Zn oxide (ZnO) is primarily added to broiler chicken feed. However, due to the low utilization rate of ZnO in poultry production, nano-ZnO has a great specific surface area and high catalytic efficiency [11]. Thus, in order to increase the inorganic Zn bioavailability, nano-ZnO opens a window for better bioavailable inorganic Zn sources.

The production of animal and poultry feed currently relies on the extensive use of nanomaterials with particle sizes ranging from 1–100 nm, which are exploited for their surface effects and distinctive physical, chemical, and biological properties. Nano-ZnO significantly increases broiler weight [12], antioxidant capacity, and cellular immune function compared to ZnO [13]. In addition, elevated levels of nano-ZnO can also improve the richness of the gut microbiota of animals, significantly changing its composition and metabolism [14]. However, the precise nano-ZnO particle size that supports the most rapid growth of broiler chicks has not been studied. Accordingly, we investigated the specific effects of adding nano-ZnO of various particle sizes to feed on the growth, immunology, morphology of the digestive tract, and composition of the gut microbiota of broilers.

## 2. Materials and Methods

### 2.1. Birds and Experimental Design

Our animal protocol was approved by the Institutional Animal Care and Use Committee of Hunan Agricultural University, China (No. 2020034). ZnO was purchased from Changsha Leyuan Biotechnology Co., Ltd. (Changsha, China), and its detected Zn content was 76%. Nano-ZnO with a particle size of 21 nm and a Zn content of 65% was purchased from Zhangjiagang Free Trade Zone Hualu Nanomaterials Co., Ltd. (Suzhou, China). Nano-ZnO with a particle size of 82 nm and a Zn content of 80% was purchased from Jiangxi Province Huaderun Fine Chemical Factory (Yichun, China).

Seventy-five one-day-old male Arbor Acres broiler chickens were randomized into three groups, including the conventional ZnO group (NC, *n* = 25), the nano-ZnO group with a mean particle size of 82 nm (ZNPL, *n* = 25), and the nano-ZnO group with a mean particle size of 21 nm (ZNPS, *n* = 25), each of which contained five replicate pens per group with five chicks per pen, in this study. Diet supplemented with either 40 mg/kg of conventional ZnO or nano-ZnO.

Food and water were provided ad libitum to all birds, and birds were maintained at an age-appropriate temperature, light regime, and humidity. The composition of the basal diets and the nutrients used were chosen to comply with the recommendations of the National Research Council (NRC, 1994; Table 1 and Table 2). On day 42, all birds were weighed and euthanized by CO_2_ inhalation to allow the collection of intestinal and blood samples, as well as lymphoid organs. As intestinal samples, duodenum, jejunum, and ileum segments were taken from midpoint with a 2 cm length. Intestinal tissues were separated, rinsed with saline, and placed on formalin. They were dehydrated, embedded in paraffin, and cut into 5-mm sections. The sections were stained with hematoxylin and eosin by mounting on glass slides and observed under an Olympus light microscope (Olympus, Tokyo, Japan) for histological examination. Villus height and crypt depth were measured by Image (Image-Pro Plus 6.1 Media Cybernetics, Rockville, MD, USA), and finally, the villus height to crypt depth ratio was calculated [15]. Serum was obtained from blood samples by centrifuging (3500× *g*, 4 °C, 10 min) and kept at −20 °C. In addition, the cecal contents were collected aseptically in sterile tubes, transferred immediately to liquid nitrogen, and kept at −80 °C for further study.

Premix provided per kilogram of diet: vitamin A, 9900 IU; vitamin D3, 3600 IU; vitamin E, 24 IU; vitamin K, 2.4 mg; vitamin B_1_, 1.8 mg; vitamin B_2_, 7.5 mg; vitamin B_6_, 3 mg; vitamin B_12_, 0.018 mg; vitamin B_7_, 0.09 mg; vitamin B_9_, 1.2 mg; nicotinamide, 36 mg; vitamin B_5_, 9.6 mg; all purchased from Shanghai Furant Animal Health Co., Ltd. (Shanghai, China).

### 2.2. Performance Parameters

During the feeding process, the initial weight, final weight, and feed intake of the chickens were recorded with accuracy, and the mean daily weight gain (ADG), mean daily feed intake (ADFI), and feed-to-gain ratio (F/G) were calculated from days 1 to 42. To determine indices of immune organs, the spleen, thymus, and bursa were collected and weighed. The lymphoid organ index was calculated according to the following formula: Lymphoid organ index (g/kg BW) = the weight of the lymphoid organ (g)/body weight (kg) [16].

### 2.3. Residual Zinc Content of Broiler Excreta

Broiler feces were collected one day prior to the end of the experiment, and residual zinc content was quantified using inductively coupled plasma optical emission spectrometry (ICP-OES) (Thermo Scientific TM iCAPTM 7000, Cambridge, UK), as described earlier [17].

### 2.4. Blood Routine and Biochemical Index Determination

On day 42, after weighing the chickens, 10 mL of blood was collected from the wing vein and stored in anticoagulant tubes with heparin. Besides, 10 mL of blood was collected from the wing vein and stored in the non-anticoagulant tubes, centrifuged (3500× *g*, 4 °C, 10 min), and the divided serum was placed into a microcentrifuge tube as a reserve. An automated mindray blood cell analyzer (BC-2800vet, Shenzhen Mindray Animal Medical Technology Co., Ltd., Shenzhen, Guangdong, China) was used to measure the number of red blood cells (RBC), white blood cells (WBC), hemoglobin (HGB), and platelets (PLT) in the whole blood sample. The activities of superoxide dismutase (SOD) and catalase (CAT), as well as the levels of glutathione (GSH), total antioxidant capacity (T-AOC), malondialdehyde (MDA), and lysozyme (LZM), were determined using commercially available assay kits in accordance with the corresponding procedures supplied by the manufacturer (Nanjing Jiancheng Institute of Bioengineering, Nanjing, China). Immunoglobulin (IgA), immunoglobulin (IgG), immunoglobulin (IgM), metallothionein (MT), and the inflammatory factors Interleukin-1β (IL-1β), Interleukin-2 (IL-2), Interleukin-6 (IL-6), Interleukin-10 (IL-10), tumor necrosis factor-α (TNF-α), and interferon-γ (IFN-γ) in the serum were determined using ELISA kits from Cusabio Biotech Co., Ltd., Houston, TX, USA.

### 2.5. Intestinal Morphology

At the time of dissection, duodenum, jejunum, and ileum of appropriate size were cut and fixed in a fixative solution of 4% neutral paraformaldehyde. Fixed tissues were embedded in paraffin and sectioned into 4 μm sections [18], followed by staining with hematoxylin and eosin. Villus height (VH) and crypt depth (CD) were measured on 8 to 10 well-oriented villi and corresponding crypts from each section of all segments of the intestine using a Nikon ECLIPSE 80i light microscope (Nikon Corporation, Tokyo, Japan), and the ratio of villi height/crypt and depth (VH/CD) was calculated [15].

### 2.6. 16S rRNA Sequencing, DNA Extraction, and PCR

Total bacterial DNA was extracted from the fecal samples using a magnetic soil and stool DNA kit (TIANGEN BIOTECH (BEIJING) CO., Ltd., Beijing, China). DNA amplification with the primer set 515F/806R (f:5′-GTGCCAGCMGCCGCGGTAA-3′, r: 5′-GGACTACHVGGGTWTCTAAT-3′), targeting the V4 regions of bacterial 16S rDNA, was performed as follows: All PCR mixtures contained 15 μL of Phusion^®^ High-fidelity PCR Master Mix (New England Biolabs), 1 μM of each primer, and 10 ng target DNA. The cycling conditions consisted of a first denaturation step at 98 °C for 1 min, followed by 30 cycles at 98 °C (10 s), 50 °C (30 s), and 72 °C (30 s), and a final 5-min extension at 72 °C. An equal volume of 1× loading buffer (containing SYB green) was mixed with PCR products and electrophoresed on 2% agarose gel for DNA detection. The PCR products were mixed in equal proportions, and a universal DNA PCR purification kit (TIANGEN BIOTECH CO., Ltd., Bei Jing, China, Catalog #: DP214) was used to purify the mixed PCR products. Following the manufacturer’s recommendations, sequencing libraries were generated using the NEB Next Ultra DNA Library Prep Kit (New England Biolabs Ltd., Ipswich, MA, USA, Catalog #: E7370L). Library quality was assessed on an Agilent 5400 system (Agilent Technologies, Inc., USA) and quantified using real-time PCR (1.5 nM). Finally, the library was sequenced on an Illumina NovaSeq platform (Illumina, Inc., San Diego, CA, USA), and 250-bp paired-end reads were generated. Paired-end reads are assigned to samples based on their unique barcode and truncated by cutting off the barcode and primer sequence. Paired-end reads are merged using FLASH (V1.2.7, http://ccb.jhu.edu/software/FLASH/, accessed on 16 January 2020), a very fast and accurate analysis tool designed to merge paired-end reads when at least some reads overlap with reads generated from the other end of the same DNA fragment, with the splice sequence referred to as the original tag. The raw tags are quality filtered under specific filtering conditions to obtain high-quality clean tags according to the QIIME (V1.7.0, http://qiime.org/index.html, accessed on 16 January 2020) quality control process. The tags are compared with a reference database (Gold database, http://drive5.com/uchime/uchime_download.html, accessed on 16 January 2020) using the UCHIME algorithm (UCHIME algorithm, (http://www.drive5.com/usearch/manual/uchime_algo.html) download.html, accessed on 16 January 2020) was compared to detect chimeric sequences, and then chimeric sequences are removed. Valid tags were then finally obtained (Novogene Bioinformatics Technology Co., Ltd., Bei Jing, China). Sequence analysis was performed by Uparse software (Uparse v7.0.1001, http://drive5.com/uparse/, accessed on 16 January 2020). Sequences with ≥97% similarity were assigned to the same OTUs. Representative sequences for each OTU were screened for further annotation. For each representative sequence, the classification information was annotated using the Silva2 database based on the RDP classifier (version 123.2, http://sourceforge.net/projects/rdpclassifier/, accessed on 16 January 2020) algorithm [19]. Shannon, Simpson, Chao1, and the ACE index were calculated in QIIME (version 1.9.1). The stacked circle plot of the OTU composition was visualized in the R project VennDiagram package (Version 3.0.3). Principal coordinates analysis (PCoA) of unweighted distances was generated and plotted in R (Version 2.15.3) WGCNA. The relative abundance of bacteria was visualized in R (Version 3.1.0). The cladogram and LEfSE assay were calculated in LEfSe (Version 1.0). Correlations between gut microbial composition (relative abundance of genera) and the BW, ADG, F:G, spleen index, thymus index, TNF-α, IFN-γ, IL-1β, IL-2, MT, and SOD-related change were evaluated with Spearman’s correlation. The correlations with a *p*-value less than 0.05 were considered statistically significant. The value range of the correlation coefficient (R-value) is −1 to 1. The closer the R value is to −1, the stronger the negative correlation is, and the closer the R value is to 1, the stronger the positive correlation is [20]. Besides, the strength of correlation during 0–0.20 was negligible, 0.21–0.35 was weak, 0.36–0.67 was moderate, 0.68–0.90 was strong, and 0.91–1.00 was very strong. The Spearman’s correlation was calculated in the R (Version 2.15.3) psych package.

### 2.7. Statistical Analysis

SPSS version 25.0 (IBM SPSS, Chicago, IL, USA) and GraphPad Prism version 9.0 (GraphPad Software, San Diego, CA, USA) were used for statistical analysis. Homogeneity of variance test and one-way ANOVA were performed for data conforming to a normal distribution, the LSD method was used for the post-test of neat variance, and tamheni T2 was used for irregularities. The test level of *p* < 0.05 was considered significant, and *p* < 0.01 was highly significant.

## 3. Results

### 3.1. Effects of Two Particle Sizes of Nano-ZnO on the Growth Performance of Broilers

As shown in Table 3, there were no relevant differences between treatment groups in the average initial body weight and ADFI. Compared with the NC group, final body weight and ADG were higher in the ZNPS (*p* < 0.05) group. Moreover, the F/G ratio of ZNPS group was decreasing than that of NC group (*p* < 0.05).

### 3.2. Effects of Two Particle Sizes of Nano-ZnO on the Immune Organ Index of Broilers

As shown in Table 4, the spleen index of ZNPS was higher than that of NC (*p* < 0.05), while the thymus index of ZNPL and ZNPS was higher than that of the NC group (*p* < 0.01). The Bursa of Fabricius Index did not differ significantly between the three groups.

### 3.3. Effects of Two Particle Sizes of Nano-ZnO on Residual Zinc in Broiler Excreta

There was no significant difference in the residual Zn content in broilers’ excreta (Figure 1).

### 3.4. Effects of Two Particle Sizes of Nano-ZnO on Blood Routine and Biochemical Indices in Broilers

Significant differences were not observed in routine blood test results and immunoglobulin content (Table 5). But the MT level in the ZNPS group was higher than that in the NC group (*p* < 0.05). In the present study, the LZM level of ZNPL and ZNPS was higher than that of NC (*p* < 0.01).

### 3.5. Effects of Two Particle Sizes of Nano-ZnO on Serum Antioxidant Indices of Broilers

As shown in Table 6, the SOD activity of the ZNPS group increased (*p* < 0.05). However, GSH, T-AOC, CAT, and MDA were not significantly different.

### 3.6. Effects of Two Particle Sizes of Nano-ZnO on Serum Inflammatory Factors in Broilers

As shown in Table 7, the levels of IL-1β and TNF-α in the ZNPS group were lower than those in the NC group (*p* < 0.05; *p* < 0.01). The IL-2 serum level in the ZNPS group was elevated when compared with the NC and the ZNPL groups (*p* < 0.01). In addition, the serum levels of IFN-γ in ZNPL and ZNPS were higher than those in the NC group (*p* < 0.01). The levels of IL-6 and IL-10 were not significantly different between the three groups (*p* > 0.05).

### 3.7. Effects of Two Particle Sizes of Nano-ZnO on Intestinal Morphology and Structure in Broilers

As shown in Figure 2 and Table 8, the ZNPS group showed elevated duodenum and ileum VH when compared with the NC group (*p* < 0.01). At the same time, the VH in the ZNPS group was higher than that in the NC group (*p* < 0.05). Moreover, the VH of the jejunum of the ZNPL group was higher than that of NC or ZNPS (*p* < 0.05). Moreover, the duodenum and ileum CD of the ZNPS group was higher than that of ZNPL (*p* < 0.01), and the VH/CD of the duodenum, jejunum, and ilenum of ZNPL and ZNPS were higher than that of NC (*p* < 0.01). At the same time, the duodenum and jejunum VH/CD were higher in the ZNPS group than in the ZNPL group (*p* < 0.01).

### 3.8. Microbial Diversity

Bacterial alpha diversity indices for the three groups are shown in Figure 3. There was no significant difference in Shannon and Simpson at OTU level among the three groups. Moreover, there was no significant difference in Chao1 and ACE in OTU (Figure 3C,D). Venn analysis showed that the OTU numbers in the NC, ZNPL, and ZNPS groups were 1079, 976, and 1227, respectively (Figure 3E). Additionally, as shown in Figure 3F, there was no obvious clustering of the samples into the three groups. These results suggest that nano-ZnO does not affect the microbiota of the chicken cecum (Figure 3F).

### 3.9. Effects of Two Particle Sizes of Nano-ZnO on the Composition of the Intestinal Flora of Broilers

The gut flora of these three groups was mainly composed of Firmicutes and Bacteroidetes (Figure 4A). On the phylum level, the abundance of Bacteroidetes in the ZNPL group decreased significantly compared with the ZNPS group (Figure 5B). However, the ratio of Firmicutes/Bacteroidetes was higher in the ZNPL than in the NC and ZNPS (*p* < 0.05 and *p* < 0.01, respectively) (Figure 5C). On the family level, the proportion of Bacteroideae in the ZNPS group was higher than that of the ZNPL (*p* < 0.01; Figure 5D). Moreover, the highest proportion of Lachnospiraceae and Erysipelotrichaceae was found in the ZNPL group (*p* < 0.05, *p* < 0.01; Figure 5E,F).

As shown in Figure 4D,E, LDA scores of four or greater were confirmed by LEfSe. On the genus level, the number of unidentified_Lachnospiraceae in the ZNPL group was increased, and the linear discriminant analysis showed that the relative abundance of Bacteroides was elevated in ZNPS. Besides, the ZNPS group possessed significantly higher levels of Bacteroides than that in the NC and ZNPL groups (*p* < 0.05, *p* < 0.01; Figure 5G). ZNPL significantly increased the relative abundances of unidentified_Lachnospiraceae and Lachnoclostridium compared with those in the NC group (Figure 5H,J). Compared with the ZNPS group, the abundance of unidentified_Lachnospiraceae, Blautia, Lachnoclostridium, unidentified_Erysipelotrichaceae, and Intestinimonas was significantly increased in the ZNPL group (Figure 5H–L).

### 3.10. Intestinal Flora of Broilers Intestinal Microbiota Correlation Analysis with Phenotypes in Broilers

As shown in Figure 6, unidentified_Lachnospiraceae was positively correlated with IFN-γ (R = 0.568, *p* = 0.027). The Blautia was positively correlated with SOD (R = 0.668, *p* = 0.006). The unidentified_Erysipelotrichaceae was positively correlated with SOD (R = 0.582, *p* = 0.023). Furthermore, Intestinimonas was negatively connected to BW and MT (R = 0.614, *p* = 0.015; R = 0.515, *p* = 0.049). However, Intestinimonas was positively related to the spleen index and IFN-γ (R = 0.547, *p* = 0.035; R = 0.609, *p* = 0.016). Compared with the ZNPS group, the abundance of Blautia, unidentified_Erysipelotrichaceae and Intestinimonas was notably increased in the ZNPL group (*p* < 0.05, *p* < 0.01; Figure 5I,K,L). Additionally, compared to the NC and ZNPS groups, the relative abundance of unidentified Lachnospiraceae was higher in the ZNPL group (*p* < 0.05, *p* < 0.01; Figure 5H).

## 4. Discussion

Zinc (Zn) can meet the nutritional requirements of broilers and plays a crucial role in their health. Nanoparticles have been widely utilized in animal husbandry to increase the usage of trace elements in animal diets as society has developed [21,22]. Research showed that nano-ZnO could improve feed conversion and growth performance [23]. In our experiments, the BW, ADG, and F/G were significantly improved in the ZNPS group compared to the NC group. These results may be attributed to the good digestion and absorption of nutrients in the gastrointestinal tract by ZNPS and the high bioavailability of the small particle size form of nano-ZnO. The ability of broilers to absorb nutrients is determined by their villus height, crypt depth, and VH/CD ratio, which form the foundation of their digestive health and function [24,25]. In our results, an increase in the intestinal villus height and VH/CD in the ZNPS group, which are positively correlated with the absorption and utilization of nutrients [26]. Besides, compared with the NC group, ZNPL and ZNPS notably increased the VH/CD of the jejunum, ileum, and duodenum, which revealed a higher absorption surface area in the intestine to increase the absorption of nutrients by the nano-ZnO of small particle size.

Studies have shown that Zn increases the activation and proliferation of lymphocytes, primarily T cells and natural killer (NK) cells, and that the immunomodulatory effects of Zn also lead to increased activity of thymocytes, macrophages, and heterophil cells, as well as increased antibody production, thereby enhancing the potential for humoral responses [27]. An increase in spleen size and weight is usually correlated to B cell differentiation and the adaptive immune system response [28]. According to the report, weight gain in broilers is positively correlated with a great immune system [29]. In our experiments, ZNPL can also increase the thymus index, and ZNPS can significantly increase both the spleen and thymus index. The results indicated that there was a significant difference in body weight in the ZNPS group when compared with the NC group. A basal diet supplemented with 80 mg/kg of ZnO and nano-ZnO resulted in significant differences between ZnO and nano-ZnO treatments in Zn deposition in excreta [30]. However, in our experiments, the fecal zinc content in the ZNPS group showed a decreasing trend, but the difference was not significant when compared with the NC group. This result may be caused by the 40 mg/kg of ZnO and nano-ZnO in the basal diet, which in our experiments was less than 80 mg/kg of ZnO and nano-ZnO. It was discovered that SOD activity was an indication for assessing an organism’s immunological function [31]. Zn acts as an activator of SOD, and high Zn bioavailability increases antioxidant enzyme activity. Studies have shown that, compared with zinc oxide, the SOD activity of the liver, pancreas, and serum can be increased by adding nano-zinc oxide into the feed [30]. The serum SOD levels of the ZNPS group were considerably higher than those of the NC group in our research, demonstrating that ZnO nanoparticles with smaller particle sizes have better effects. In the present study, the results of the blood biochemical analyses showed that the blood LZM concentrations in the ZNPL and ZNPS groups were significantly higher than those in the NC group. LZM is not only a bacteriostatic and anti-inflammatory protein but also plays a significant role in maintaining homeostasis in normal defense function and non-specific immunity [32]. Dietary nano-ZnO showed that it increased Zn absorption and deposition via enhancing the expression of transporters (MT) in the jejunum [33]. In our experiments, the MT blood level was higher in the ZNPS group when compared to the NC group. MT is an important maintainer of the Zn pool of the organisms that MT can scavenge free radicals, reduce blood viscosity, improve the detoxification ability of heavy metals [34,35], and ultimately enhance immunity. In addition, our results also revealed that ZNPS significantly decreased production of the inflammatory mediators TNF-α and IL-1β [36].

It has been extensively suggested that the intestinal microbiota plays an important role in the maintenance of intestinal health and growth promotion [37,38,39]. Besides, the complex gut microbiome affects immune function and gut barrier function [40,41,42]. At present, the studies of nano-ZnO on the intestinal flora of broiler chickens are few. However, some studies have shown that dietary supplementation with nano-zinc oxide can increase the abundance of beneficial bacteria in animals [14,43]. The results of the current investigation showed that broilers’ alpha diversity of cecal microflora was unaffected by the supplementation of ZNPL and ZNPS. The diversity study further demonstrated that ZNPL and ZNPS had not altered the primary gut flora in the cecum of broilers. It suggested that the number of species and diversity of the cecal microbial community were unaffected by ZNPL and ZNPS. According to our findings, Firmicutes and Bacteroidetes made up the majority of the cecal bacteria in broilers, which was consistent with earlier research [44,45]. The Firmicutes and Bacteroidetes were involved in the production and metabolism of energy, especially starch digestion and microbial fermentation [45,46,47]. Taxonomic profiling demonstrated that ZNPL treatment was able to decrease the abundance of Bacteroidetes as well as increase the Firmicutes/Bacteroidetes ratio in our experiment. Studies have reported that the promoted growth performance in broilers is frequently correlated with an elevated Firmicutes/Bacteroidetes ratio [48]. Our results indicated that ZNPL treatment in the diet could increase the BW and ADG of broilers, but there was no statistical difference when compared with the NC group. The relative abundance of the family Lachnospiraceae and the genus unidentified_Lachnospiraceae increased in the ZNPL group. According to numerous research studies, the Lachnospiraceae family is related to the gut health of broilers. The high abundance of beneficial bacteria can help lower the number of dangerous bacteria in the intestine, further promoting the immune response [42,49]. The proportion of Erysipelotrichaceae was also increased in ZNPL at the family level, which was reported to be beneficial to normal glucose metabolism and alleviate insulin resistance [50,51]. In addition, ZNPL and ZNPS also altered the relative abundance of some taxa, such as Bacteroides, Lachnoclostridium, Intestinimonas, and the genus Blautia, in the present study. We found that the proportion of Bacteroides genus was higher than 20% among all the groups. Interestingly, we also discovered that ZNPS administration enhanced the proportion of the Bacteroides genus. Several studies have shown that Bacteroides play a vital role in the fermentation of carbohydrates to produce SCFAs, participate in polysaccharide metabolism, mediate colonization resistance to invading enteric pathogens, and regulate immunity [40,52,53]. Meanwhile, our results showed that the VH/CD and immune organ indexes were increased in the ZNPS group. Therefore, ZNPS might play a significant role in improving nutrient absorption and gut health in our study. It has been reported that Lachnospiraceae and Intestinimonas are crucial producers of butyrate residing in the gut microbiota [54,55]. Butyrate provides energy to intestinal epithelial cells as well as playing an important role in the inhibition of inflammation and the promotion of intestinal development [56,57,58]. The Spearman correlation analysis in this study showed that the relative abundance of Intestinimonas was positively correlated with the spleen index and IFN-γ of broilers (R = 0.547, *p* = 0.035; R = 0.609, *p* = 0.016). Besides, the relative abundance of Blautia was positively correlated with the SOD (R = 0.668, *p* = 0.006). Research has shown that Blautia can degrade different types of carbohydrates and produce metabolites such as acetic acid and lactic acid [59]. The increase in the content of these metabolites is linked to the host’s gut health [60], which may provide energy to the body and reduce inflammation.

Collectively, ZNPL and ZNPS may enhance immunity and enhance antioxidant capacity in broilers, while ZNPS may provide birds with improved growth performance relative to ZNPL. ZNPL and ZNPS may also increase SCFA-producing bacteria that inhibit the proliferation of harmful bacteria so as to provide a healthier gut microecology. The results of this study suggested that ZNPL and ZNPS in diets had the potential to improve growth performance, immunity, and intestinal health in broiler chickens.

## 5. Conclusions

Nano-ZnO can improve the growth performance of broiler chickens, promote the development of immune organs, improve the non-specific immunity of the body, and have certain anti-inflammatory effects, and the above improvement effects are negatively correlated with the particle size of nanoparticles. The particle size of nano-ZnO may affect the bioavailability of zinc in broilers. In addition, nano-ZnO can also enrich some of the beneficial intestinal bacteria.

## Figures and Tables

**Figure 1 animals-13-01454-f001:**
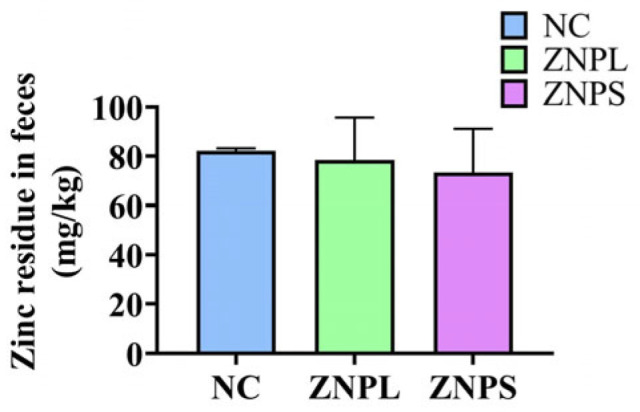
Effects of two particle sizes of nano zinc oxide on residual zinc in broiler excreta. Values are expressed as means with SD.

**Figure 2 animals-13-01454-f002:**
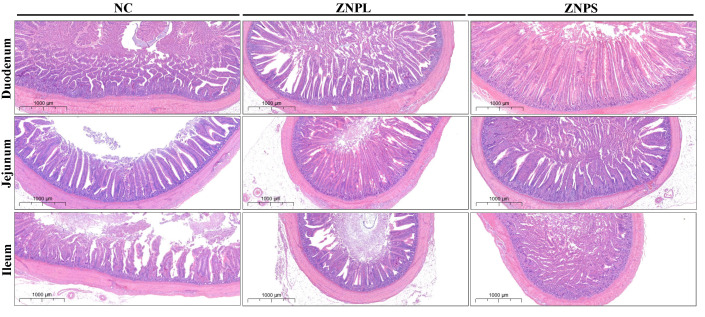
The representative histologic photomicrograph for the intestinal tissues in different groups.

**Figure 3 animals-13-01454-f003:**
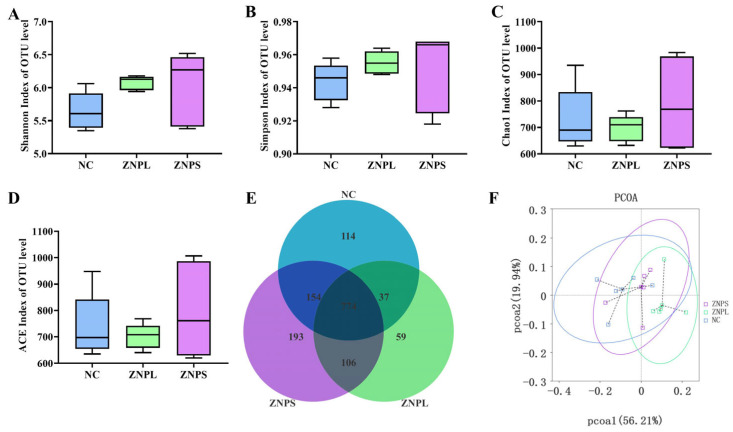
Microbial Alpha Diversity Index in Broiler Cecum. (**A**) Shannon index at OTU level; (**B**) Simpson index at OTU level; (**C**) Chao1 index at OTU level; (**D**) ACE index at OTU level; (**E**) Venn diagrams showing the unique and common OTUs in gut microbiota; (**F**) PCoA of OTUs.

**Figure 4 animals-13-01454-f004:**
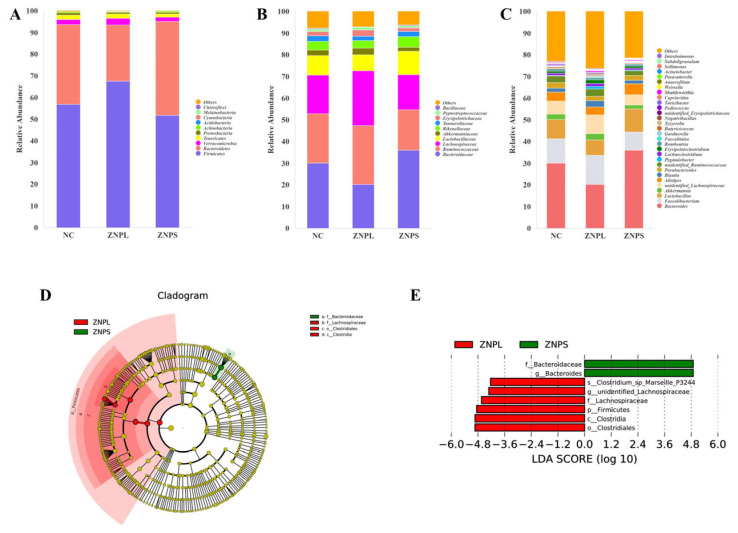
The composition of gut microbiota in broilers. (**A**) Percentage of community abundance at the phylum level. (**B**) Percentage of community abundance at the family level. (**C**) Percentage of community abundance at the genus level. (**D**) Cladoram from the LEfSe assay. (**E**) Generation of the LDA score (LDA > 4, *p* < 0.05) for differential abundance of microorganisms.

**Figure 5 animals-13-01454-f005:**
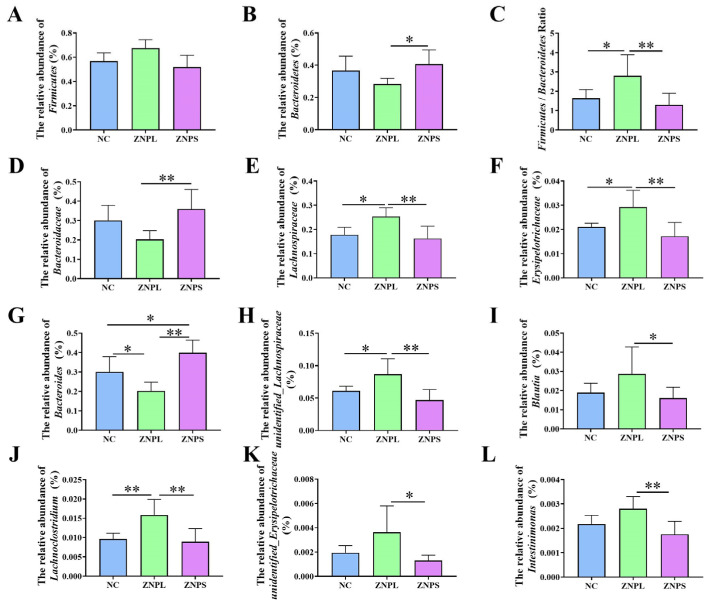
Differential bacteria in broilers at the phylum, family, and genus levels. (**A**) Relative abundance of Firmicutes at the phylum level. (**B**) Relative abundance of Bacteroidetes at the phylum level. (**C**) Firmicutes/Bacteroidetes ratio. (**D**–**F**) Relative abundance of Bacteroidaceae, Lachnospiraceae, and Erysipelotrichaceae at the family level. (**G**–**L**) Relative abundance of six representative genera. * is significantly different (*p* < 0.05). ** are significantly different (*p* < 0.01).

**Figure 6 animals-13-01454-f006:**
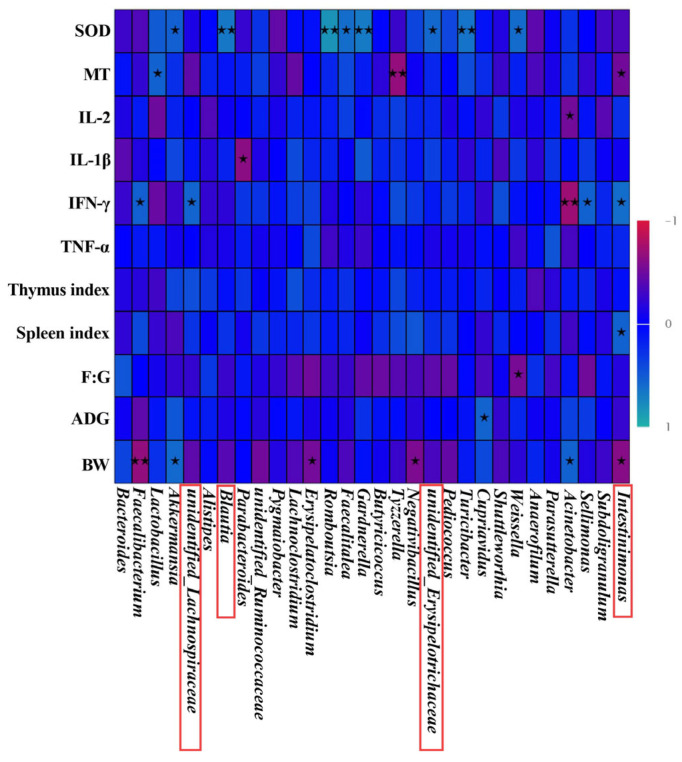
Predictive analysis of intestinal microbiota phenotypes in broilers. Heatmap of Spearman’s correlation between gut microbial composition (relative abundance of genera) and indexes related to BW, ADG, F:G, spleen index, thymus index, TNF-α, IFN-γ, IL-1β, IL-2, MT, and SOD in broilers. * is significantly different (*p* < 0.05). ** are significantly different (*p* < 0.01); The red boxes represent significant differences in the relative abundance of the bacteria between the three groups; The colors range from nattier blue (positive correlation) to red (negative correlation).

**Table 1 animals-13-01454-t001:** Composition of experimental diets for broilers from days 0–21 and days 22–42.

Ingredients (%, as Fed)	0–21 Days Content %	22–42 Days Content %
Corn	57	61.5
Soybean meal	31	24.5
Fish meal	4	5
Oil	4	5
CaHPO_4_	1.47	1.37
Limestone	0.9	1
L-lys·HCl	0.25	0.3
DL-methionine	0.2	0.15
L-Threonine	0.15	0.15
Choline chloride	0.2	0.2
NaCl	0.3	0.3
Multi-vitamin1	0.03	0.03
Premix1	0.5	0.5

**Table 2 animals-13-01454-t002:** Composition of the mineral premix.

Ingredients (%)	Effective Concentration (EC, %)	ZnO (%)	Nano-ZnO (%)	
			82 nm	21 nm
FeSO_4_·H_2_O	0.376	5.4	5.4	5.4
CuSO_4_·5H_2_O	0.25	0.64	0.64	0.64
ZnO	0.76	1.05	-	-
nano-ZnO	0.65	-	1.22	-
nano-ZnO	0.8	-	-	1
MnSO_4_·H_2_O	0.318	7.6	7.6	7.6
Ca(IO_3_)_2_	Iodine 1%	1.4	1.4	1.4
Oil	1	2.26	2.26	2.26
Na_2_SeO_3_	Selenium 1%	0.6	0.6	0.6
Maifanstone	1	81.05	80.88	81.1

Notes: This mineral premix was custom-made without the addition of ZnO.

**Table 3 animals-13-01454-t003:** Effects of two particle sizes of nano zinc oxide on the growth performance of broilers from days 1 to 42.

Item	NC ^1^	ZNPL ^1^	ZNPS ^1^	SEM ^2^	*p*-Value
D 1 BW (g)	41.02	40.86	40.66	0.12	0.247
D 42 BW (g)	1550.20 ^b^	1639.27 ^a, b^	1786.90 ^a^	48.23	0.048
ADG (g/day)	35.93 ^b^	38.06 ^a, b^	41.58 ^a^	1.15	0.048
ADFI (g/day)	65.35	60.83	63.01	2.14	0.679
F/G	1.82 ^a^	1.60 ^a, b^	1.52 ^b^	0.06	0.038

^1^ NC = conventional ZnO; ZNPL = nano-ZnO with a mean particle size of 82 nm; ZNPS = nano-ZnO with a mean particle size of 21 nm; ^2^ SEM = standard error of the mean; ^a, b^—means with different superscripts within a row are significantly different at *p* < 0.05. Values are expressed as means with SEM. Abbreviations: BW, body weight; ADG, average daily gain; ADFI, average daily feed intake; F/G, feed:gain ratio.

**Table 4 animals-13-01454-t004:** Effects of two particle sizes of nano zinc oxide on the immune organ index of broilers.

Item	NC ^1^	ZNPL ^1^	ZNPS ^1^	SEM ^2^	*p*-Value
Spleen index (g/kg BW)	1.33 ^b^	1.43 ^a, b^	1.73 ^a^	0.07	0.020
Thymus index (g/kg BW)	5.11 ^b^	7.12 ^a^	8.82 ^a^	0.41	0.001
Bursa of Fabricius index (g/kg BW)	2.53	2.11	2.82	0.19	0.540

^1^ NC = conventional ZnO; ZNPL = nano-ZnO with a mean particle size of 82 nm; ZNPS = nano-ZnO with a mean particle size of 21 nm; ^2^ SEM = standard error of the mean; ^a, b^—means with different superscripts within a row are significantly different at *p* < 0.05. Values are expressed as means with SEM.

**Table 5 animals-13-01454-t005:** Effects of two particle sizes of nano-ZnO on blood routine and biochemical indices in broilers.

Item	NC ^1^	ZNPL ^1^	ZNPS ^1^	SEM ^2^	*p*-Value
WBC	240.35	237.72	238.59	0.76	0.354
RBC	2.60	2.48	2.60	0.30	0.945
PLT	14.83	11.17	12.17	0.76	0.150
HGB	141.25	136.50	139.58	1.15	0.551
IgA	60.90	70.89	70.44	3.66	0.278
IgG	5.94	5.74	5.74	0.43	0.858
IgM	5.27	7.17	5.13	0.43	0.898
MT	896.72 ^b^	960.01 ^a, b^	1072.98 ^a^	35.04	0.041
LZM	118.23 ^b^	144.68 ^a^	148.84 ^a^	4.91	0.008

^1^ NC = conventional ZnO; ZNPL = nano-ZnO with a mean particle size of 82 nm; ZNPS = nano-ZnO with a mean particle size of 21 nm; ^2^ SEM = standard error of the mean; ^a, b^—means with different superscripts within a row are significantly different at *p* < 0.05. Values are expressed as means with SEM. WBC, white blood cells; RBC, red blood cells; PLT, platelets; HGB, hemoglobin; IgA, immunoglobulin A; IgG, immunoglobulin G; IgM, immunoglobulin M; MT, metallothionein M; and LZM, lysozyme.

**Table 6 animals-13-01454-t006:** Effects of two particle sizes of nano-ZnO on the serum antioxidant indices of broilers.

Item	NC ^1^	ZNPL ^1^	ZNPS ^1^	SEM ^2^	*p*-Value
GSH (U/mL)	31.24	57.90	49.25	6.13	0.229
SOD (U/mL)	117.25 ^b^	121.77 ^a, b^	175.33 ^a^	11.50	0.036
T-AOC (U/mL)	1.11	1.07	1.08	0.03	0.676
CAT (U/mL)	17.93	16.36	17.55	0.62	0.806
MDA (nmol/mL)	3.86	4.05	3.83	0.21	0.956

^1^ NC = conventional ZnO; ZNPL = nano-ZnO with a mean particle size of 82 nm; ZNPS = nano-ZnO with a mean particle size of 21 nm; ^2^ SEM = standard error of the mean; ^a, b^—means with different superscripts within a row are significantly different at *p* < 0.05. Values are expressed as means with SEM. GSH, glutathione; SOD, superoxide dismutase; T-AOC, total antioxidant capacity; CAT, catalase; MDA, malondialdehyde.

**Table 7 animals-13-01454-t007:** Effects of two particle sizes of nano zinc oxide on serum inflammatory factors in broilers.

Item	NC ^1^	ZNPL ^1^	ZNPS ^1^	SEM ^2^	*p*-Value
IL-1β (pg/mL)	65.70 ^a^	59.05 ^a, b^	39.41 ^b^	5.32	0.044
IL-2 (pg/mL)	0.1532 ^b^	0.1565 ^b^	0.163 ^a^	0.001	0.003
IL-6 (pg/mL)	10.99	9.82	9.74	0.61	0.414
IL-10 (pg/mL)	1.47	1.40	1.47	0.07	0.962
TNF-α (pg/mL)	171.90 ^a^	152.64 ^a^	137.31 ^b^	4.11	0.001
IFN-γ (pg/mL)	78.76 ^b^	155.50 ^a^	128.42 ^a^	8.07	0.003

^1^ NC = conventional ZnO; ZNPL = nano-ZnO with a mean particle size of 82 nm; ZNPS = nano-ZnO with a mean particle size of 21 nm; ^2^ SEM = standard error of the mean; ^a, b^—means with different superscripts within a row are significantly different at *p* < 0.05. Values are expressed as means with SEM. IL-1β, interleukin 1β; IL-2, interleukin 2; IL-6, interleukin 6; IL-10, interleukin 10; TNF-α, tumor necrosis factor-α; IFN-γ, interferon-γ.

**Table 8 animals-13-01454-t008:** Effects of two particle sizes of nano-ZnO on the intestinal morphometry of broilers.

Item Duodenum	NC ^1^	ZNPL ^1^	ZNPS ^1^	SEM ^2^	*p*-Value
VH (μm)	737.24 ^b^	584.72 ^c^	1273.38 ^a^	79.57	0.001
CD (μm)	148.80 ^a^	80.06 ^b^	135.58 ^a^	6.88	0.001
VH/CD	5.00 ^c^	7.46 ^b^	9.57 ^a^	0.62	0.001
Jejunum					
VH (μm)	666.42 ^c^	974.80 ^a^	838.98 ^b^	35.54	0.030
CD (μm)	155.22 ^a^	143.56 ^a^	100.26 ^b^	8.23	0.002
VH/CD	4.42 ^c^	6.86 ^b^	8.62 ^a^	0.53	0.001
Ileum					
VH (μm)	551.38 ^c^	800.40 ^b^	1062.32 ^a^	49.22	0.001
CD (μm)	105.16 ^b^	103.66 ^b^	132.90 ^a^	2.71	0.005
VH/CD	5.25 ^b^	7.77 ^a^	8.16 ^a^	0.42	0.001

^1^ NC = conventional ZnO; ZNPL = nano-ZnO with a mean particle size of 82 nm; ZNPS = nano-ZnO with a mean particle size of 21 nm; ^2^ SEM = standard error of the mean; ^a, b, c^—means with different superscripts within a row are significantly different at *p* < 0.05. Values are expressed as means with SEM. VH, villus height; CD, crypt depth; VH/CD, villus height:crypt depth ratio.

## Data Availability

The data presented in this study are available on reasonable request from the corresponding author.

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
