# Peer review of "Effect of Two Particle Sizes of Nano Zinc Oxide on Growth Performance, Immune Function, Digestive Tract Morphology, and Intestinal Microbiota Composition in Broilers"

_animals, 2023, doi:10.3390/ani13091454_

Round 1

Reviewer 1 Report

The manuscript entitled "Effect of two particle sizes of nano zinc oxide on growth perfor- 2 mance, immune function, digestive tract morphology and intes- 3 tinal microbiota composition in broilers" is interesting. some modifications are needed. 

The whole manuscript should be revised for language by professional language editing service. Many grammatical and structural errors are present (ex... line 101, contents of cecal were collected--- it should be cecal contents or contents of cecum). Some sentences are incomplete (ex.. line 48, Zn in the form of ions or chemical compounds in the body's organs contain.... I cant understand what did you mean!!!

Line 138: add reference for the used technique.

Line 143: 2.616. S RNA sequencing, DNA extraction and PCR (correct numbering)

Line 228: I suggest to add representative histologic photomicrograph for the intestinal tissues in different groups

figures 3,4,5... too small size, it was so difficult to identify their contained data... add high resolution figures 

Author Response

Dear Reviewer:

We are grateful for your valuable time in making the constructive remarks. Based on your valuable advice, we have amended the relevant part of the manuscript and answered your questions below.

Point 1: The whole manuscript should be revised for language by professional language editing service. Many grammatical and structural errors are present (ex... line 101, contents of cecal were collected--- it should be cecal contents or contents of cecum). Some sentences are incomplete (ex.. line 48, Zn in the form of ions or chemical compounds in the body's organs contain.... I cant understand what did you mean!!!.

Response 1: Thank you for your comment. Our manuscript has been edited to ensure language and grammar accuracy under your advice. Editing was performed by professional editors at Editage, a division of Cactus Communications. Besides, we apologize for our mistakes to write the sentence in lines 48-49, “Zn in the form of ions or chemical compounds in the body's organs contain.” So, in lines 49-50, “Zn in the form of ions or chemical compounds in the body's organs contain. By taking part in the manufacture of essential proteins, it is utilized to control a number of physiological processes in animals, including gene expression, reproduction, cell division, wound healing, and bone calcification [2].” was changed to “Zn, a mineral, is important for birds' metabolism of nutrients, appetite regulation, and cleaning up reactive oxygen species [2].”

[2] ElKatcha M, Soltan M, Elbadry M: Effect of Dietary Replacement of Inorganic Zinc by Organic or Nanoparticles Sources on Growth Performance, Immune Response and Intestinal Histopathology of Broiler Chicken. Alexandria Journal of Veterinary Sciences 2017, 55.

Point 2: Line 138: add reference for the used technique.

Response 2: Thank you for your advice. We have already added reference for the used technique in line 159, “Fixed tissues were embedded in paraffin and sectioned into 4 μm sections [18], followed by staining with hematoxylin and eosin.”

[18] Zhang Q, Zhang K, Wang J, Bai S, Zeng Q, Peng H, Zhang B, Xuan Y, Ding X: Effects of coated sodium butyrate on performance, egg quality, nutrient digestibility, and intestinal health of laying hens. Poult Sci 2022, 101:102020.

Point 3: Line 143: 2.616. S RNA sequencing, DNA extraction and PCR (correct numbering).

Response 3: We apologize for our mistakes and appreciate your comment. We have corrected the errors that you pointed out and have carefully checked the full text. Line 164: “2.616. S RNA sequencing, DNA extraction and PCR” was changed to “16S rRNA sequencing, DNA extraction and PCR”

Point 4: Line 228: I suggest to add representative histologic photomicrograph for the intestinal tissues in different groups.

Response 4: Thank you for your advice. We have already added the representative histologic photomicrograph for the intestinal tissues in different groups in Section 3.7. Besides, we have re-uploaded the high resolution figures with Figures 3, 4, and 5.

Thank you very much for your kind advice.

Reviewer 2 Report

Qu et al. investigated the effects of two nano-ZnO particle sizes in broiler chickens. The potential use of nanoparticles to improve mineral absorption and bioavailability is intriguing. However, I have some concerns, which are listed below.

Introduction

The second paragraph lacks an explanation of the research problem and the disadvantages of using conventional Zn. The authors should elaborate on this and how nanotechnology can address these issues.

Methodology

Please specify in the method whether this mineral premix was custom-made without the addition of ZnO or not?.

Line 110: change to mineral premix rather than Premix.

Line 115-116: Please specify how the relative weight of lymphoid organs was calculated.

Line 143: Correct the subheading 16S rRNA

Section 3.8: Change OUT to OTU. Check and correct all.

Discussion

The findings for fecal Zn were not discussed by the authors. Additionally, according to their study, there was no significant difference in Zn excretion between the groups, so it was illogical to claim that nano-zinc provided high bioavailability. Since the same dosage of ZnO and Nano-Zn showed comparable Zn excretion.

The authors should also discuss about the biological fate of nano-zn in gastric fluid. Because ZnO is highly soluble in gastric fluid (acidic condition), regardless of the size of nano-zn used in this study, the majority of the particles will dissociate into ions. Could the authors comment on this?

The authors should explain how Zn improves intestinal morphology and briefly explain how Zn aids in the development of lymphoid organs.

Line 313: Bursa of Fabricius rather than fabricus. Check and correct all.

Please try to connect the increased SOD enzymes in broiler serum to the high Zn bioavailability. Because Zn acts as an activator of SOD, high Zn bioavailability increases antioxidant enzyme activity. Also, discuss how Zn aids in the increase of MT levels.

One of the major concerns of using nano-zn in poultry feed is the detrimental effect on commensal bacteria. The author might want to briefly discuss the effect of nano-zn on beneficial bacteria.

Conclusion

Please revise the conclusion. Does size affect Zn bioavailability?

Author Response

Dear Reviewer:

We appreciated all valuable comments from you on our manuscript entitled “Effect of two particle sizes of nano zinc oxide on growth performance, immune function, digestive tract morphology and intestinal microbiota composition in broilers”. Based on your valuable advice, we have amended the relevant part of the manuscript and answered your questions below.

Point 1: Introduction

The second paragraph lacks an explanation of the research problem and the disadvantages of using conventional Zn. The authors should elaborate on this and how nanotechnology can address these issues.

Response 1: Thank you for your advice. We have already added the disadvantages of using conventional ZnO and the advantages of using nano-ZnO in lines 69-71, “However, due to the low utilization rate of ZnO in poultry production, nano-ZnO has great specific surface area and high catalytic efficiency[11]”.

[11] Hussan F, Krishna D, Preetam VC, Reddy PB, Gurram S: Dietary Supplementation of Nano Zinc Oxide on Performance, Carcass, Serum and Meat Quality Parameters of Commercial Broilers. Biol Trace Elem Res 2022, 200:348-353.

Point 2: Methodology

Please specify in the method whether this mineral premix was custom-made without the addition of ZnO or not?.

Response 2: Thank you for your comment. We have already added the specify in the method in line 125, “Notes: this mineral premix was custom-made without the addition of ZnO.”

Point 3: Line 110: change to mineral premix rather than Premix.

Response 3: We apologize for our mistakes and appreciate your comment. We have corrected the error, “Premix” was changed to “mineral premix”.

Point 4: Line 115-116: Please specify how the relative weight of lymphoid organs was calculated.

Response 4: Thank you for your comment. We have already added the calculation method in line 131-133, “The lymphoid organ index was calculated according to the following formula: lymphoid organ index (g/ kg BW) = the weight of lymphoid organ (g)/body weight (kg) [16].”

[16] Wang Q, Wang XF, Xing T, Li JL, Zhu XD, Zhang L, Gao F: The combined impact of xylo-oligosaccharides and gamma-irradiated astragalus polysaccharides on the immune response, antioxidant capacity, and intestinal microbiota composition of broilers. Poult Sci 2022, 101:101996.

Point 5: Line 143: Correct the subheading 16S rRNA.

Response 5: We apologize for our mistakes and appreciate your comment. We have corrected the errors that you pointed out and have carefully checked the full text. Line 164: “2.616. S RNA sequencing, DNA extraction and PCR” was changed to “16S rRNA sequencing, DNA extraction and PCR”.

Point 6: Section 3.8: Change OUT to OTU. Check and correct all.

Response 6: We apologize for our mistakes and appreciate your comment. We have corrected the errors that you pointed out and have carefully checked the full text.

Point 7: Discussion

The findings for fecal Zn were not discussed by the authors. Additionally, according to their study, there was no significant difference in Zn excretion between the groups, so it was illogical to claim that nano-zinc provided high bioavailability. Since the same dosage of ZnO and Nano-Zn showed comparable Zn excretion.

Response 7: We really appreciate your valuable comments on our manuscript. According to your advice, we have added the sentence “A basal diet supplemented with 80 mg/kg of ZnO and nano-ZnO, there were significant differences between ZnO and nano-ZnO treatments in Zn deposition in excreta [28]. However, in our experiments, fecal zinc content in ZNPS group showed a decreasing trend, but the difference was not significant when compared with NC group. This result may be caused by 40mg/kg of ZnO and nano-ZnO in the basal diet in our experiments less than 80 mg/kg of ZnO and nano-ZnO.” in lines 382-387.

[28] Abedini M, Shariatmadari F, Karimi Torshizi MA, Ahmadi H: Effects of zinc oxide nanoparticles on the egg quality, immune response, zinc retention, and blood parameters of laying hens in the late phase of production. J Anim Physiol Anim Nutr (Berl) 2018, 102:736-745.

Point 8: The authors should also discuss about the biological fate of nano-zn in gastric fluid. Because ZnO is highly soluble in gastric fluid (acidic condition), regardless of the size of nano-zn used in this study, the majority of the particles will dissociate into ions. Could the authors comment on this?

Response 8: We gratefully thank you for your precious time in making constructive remarks. During the experiment, we did neglect to detect nano-zn in gastric fluid. Thanks for your advice again, it is very important for us. We will work in the future according to your advice to improve the level of scientific research, and achieve more results. Thank you again for your constructive comments.

Point 9: The authors should explain how Zn improves intestinal morphology and briefly explain how Zn aids in the development of lymphoid organs.

Response 9: We gratefully appreciate your valuable comment. it is very important, because of your advice, let us have a great harvest. In order to make the article more precise, " ZNPS could significantly improve the morphology of jejunum, ileum, and duodenum" has been changed to " A increase in the intestinal villus height and VH/CD in ZNPS group" in lines 363 according to the comments. Besides, we have added

“Studies have shown that Zn increases the activation and proliferation of lymphocytes, primarily T cells and natural killer (NK) cells, and that the immunomodulatory effects of Zn also lead to increased activity of thymocytes, macrophages and heterophil cells, as well as increased antibody production, thereby enhancing the potential for humoral responses [25].” in lines 369-373 for explain how Zn aids in the development of lymphoid organs.

[25] Jarosz L, Marek A, Gradzki Z, Laskowska E, Kwiecien M: Effect of Zinc Sulfate and Zinc Glycine Chelate on Concentrations of Acute Phase Proteins in Chicken Serum and Liver Tissue. Biol Trace Elem Res 2019, 187:258-272.

Point 10: Line 313: Bursa of Fabricius rather than fabricus. Check and correct all.

Response 10: We apologize for our mistakes and appreciate your comment. We have corrected the errors that you pointed out and have carefully checked the full text.

Point 11: Please try to connect the increased SOD enzymes in broiler serum to the high Zn bioavailability. Because Zn acts as an activator of SOD, high Zn bioavailability increases antioxidant enzyme activity. Also, discuss how Zn aids in the increase of MT levels.

Response 11: Thanks for your suggestion. In order to connect the increased SOD enzymes in broiler serum to the high Zn bioavailability, “Dietary nano-ZnO has previously been shown to be capable of enhancing SOD activity in the liver, pancreas, and serum” has been changed to “Zn acts as an activator of SOD, high Zn bioavailability increases antioxidant enzyme activity. Studies have shown that, compared with zinc oxide, SOD activity of liver, pancreas and serum can be increased by adding nano-zinc oxide into the feed” in lines 389-392. Besides, we have added the sentence “Dietary Nano-ZnO showed it increased the Zn absorption and deposition via enhancing the expression of transporters (MT) in the jejunum [31].” in lines 401-403 to discuss how Zn aids in the increase of MT levels.

[31] Zhou B, Li J, Zhang J, Liu H, Chen S, He Y, Wang T, Wang C: Effects of Long-Term Dietary Zinc Oxide Nanoparticle on Liver Function, Deposition, and Absorption of Trace Minerals in Intrauterine Growth Retardation Pigs. Biol Trace Elem Res 2022.

Point 12: One of the major concerns of using nano-zn in poultry feed is the detrimental effect on commensal bacteria. The author might want to briefly discuss the effect of nano-zn on beneficial bacteria.

Response 12: Thank you for your comment. According to your advice, we have added “At present, the study of nano-ZnO on intestinal flora of broiler chickens is few. How-ever, some studies have shown that dietary supplementation of nano-zinc oxide can increase the abundance of beneficial bacteria in animals [14, 41].” in lines 411-414.

[14] Liu H, Bai M, Xu K, Zhou J, Zhang X, Yu R, Huang R, Yin Y: Effects of different concentrations of coated nano zinc oxide material on fecal bacterial composition and intestinal barrier in weaned piglets. J Sci Food Agric 2021, 101:735-745.

[41] Zhang H, Guan W, Li L, Guo D, Zhang X, Guan J, Luo R, Zheng S, Fu J, Cheng Y, He Q: Dietary carbon loaded with nano-ZnO alters the gut microbiota community to mediate bile acid metabolism and potentiate intestinal immune function in fattening beef cattle. BMC Vet Res 2022, 18:425.

Point 13: Conclusion

Please revise the conclusion. Does size affect Zn bioavailability?

Response 13: Thanks for your advice. According to your advice, we have added “The particle size of nano-ZnO may affect the bioavailability of zinc in broilers.” in lines 465-466.

Thank you very much for your kind advice.

Reviewer 3 Report

General comments:

I think the work presented in the paper is relevant for the field and can be beneficial to poultry production. Their results showed the small size of nano-ZnO has a significant effect in improving the immune system and at the same time improves feed conversion. These results can be used in poultry production to improve the health status of the birds and at the same time can help to prevent and control the spread of foodborne pathogen diseases caused by human poultry product consumption.

The paper is well-written; I recommend this manuscript is to be accepted with minor revisions.

In section 2.3 (“Residual Zinc content of broiler excreta”) describes the residual Zinc was only measured in the feces on the day prior to the end of the experiment. Would not be more beneficial if this measurement was done at several time points during the ZnO supplementation?

In section 2.4 (“Blood routine and biochemical indexes determination”) for the whole blood sample analysis what kind of anticoagulant was used? Was it used different techniques or sets of tubes to collect the blood for the whole blood analysis and for the serum analysis? Please, further describe the blood collection (was heparin used in the syringe for blood collection? What anticoagulant was used? Were blood samples collected in blood tubes for serum extraction?).

In section 2.4 (“Intestinal morphology”) please describe the washing/rinse steps before fixation previously cited in section 2.1 lines 98-100.

The authors show an increase in body weight and some lymphoid organs weight, concluding that the ZNPS group presented improvement in feed conversion and immune system response. In addition, ZNPS animals presented an increase in the size of the villa and crypta of all sections of the intestines, which could be correlated to an increase in the weight of the intestines of these birds. Meaning that the animals presented an increase in several organs' weight that could explain the increase in body weight. Was the carcass weight of the broiler chicken measured? If it was it is recommended to be added to the manuscript to back up the conclusion of improvement of feed conversion and growth performance. Without the carcass weight or other cut weight (breast meat, thighs, leg, or wing) it is hard to evaluate if the increase in body weight was because of growth development or just secondary to the increase in weight of internal organs.

Specific comments:

Line 97-98: “collection of the intestinal sample, and blood and lymphoid organs were harvested” The sentence sounds confusing. Suggested change to: “collection of intestinal and blood samples, and lymphoid organs.”

Line 98-100: “In addition, duodenum, jejunum, and ileum segments 2 cm in length, taken from the midpoint, were separated, rinsed with saline water, and fixed in formalin for histologic study” I suggest this sentence should be split into two sentences to improve clarity: As intestinal samples, it was collected duodenum, jejunum, and ileum segments taken from midpoint with 2 cm in length. Intestinal tissues were separated, rinsed with saline, and fixed in formalin for histologic study.

Table 3 - D42 BW (g) line: Statistical significance letters are 2b, b, and a, the legend does not describe the meaning of “2b”. Is it a typo in the table?

Section 3.5: In the text and the title of the table is referred to as “Table 4”, but it should be ”Table 6” according to the order in the manuscript.

Section 3.6: In the text and the title of the table is referred to as “Table 5”, but it should be ”Table 7” according to the order in the manuscript.

Section 3.7: In the text and the title of the table is referred to as “Table 6”, but it should be ”Table 8” according to the order in the manuscript. In addition, the table should be mice to be between the table title and the legend.

Line 303: “NC grouop” substitute for “NC group”.

Line 313: “In broiler fabricus, the spleen…” please, check the spelling. The sentence is confusing, I am not sure what the authors wanted to say.

Line 315: “splenic weight is associated with immune cell proliferation.” An increase in spleen size and weight is usually correlated to B cell differentiation and adaptive immune system response. Most immune cell proliferation (including heterophils, macrophages, and B cell progenitors) occurs in the bone marrow. Please, correct it in the text.

Line 344: “he Firmicutes and Bacteroidetes…” should be substituted for “The Firmicutes and Bacteroidetes”.

Section 5: The last sentence is missing the final period.

Author Response

Dear Reviewer:

We appreciated all valuable comments from you on our manuscript entitled “Effect of two particle sizes of nano zinc oxide on growth performance, immune function, digestive tract morphology and intestinal microbiota composition in broilers”. Based on your valuable advice, we have amended the relevant part of the manuscript and answered your questions below.

Point 1: In section 2.3 (“Residual Zinc content of broiler excreta”) describes the residual Zinc was only measured in the feces on the day prior to the end of the experiment. Would not be more beneficial if this measurement was done at several time points during the ZnO supplementation?

Response 1: We gratefully thank you for your precious time in making constructive remarks. During the experiment, we did neglect to detect the residual Zinc was only measured in the feces on the day prior to the end of the experiment. Thanks for your advice again, it is very important for us. We will work in the future according to your advice to improve the level of scientific research, and achieve more results. Thank you again for your constructive comments.

Point 2: In section 2.4 (“Blood routine and biochemical indexes determination”) for the whole blood sample analysis what kind of anticoagulant was used? Was it used different techniques or sets of tubes to collect the blood for the whole blood analysis and for the serum analysis? Please, further describe the blood collection (was heparin used in the syringe for blood collection? What anticoagulant was used? Were blood samples collected in blood tubes for serum extraction?).

Response 2: We gratefully appreciate for your valuable comment. According to your advice, we have added “On day 42, After weighing the chickens, 10mL blood was collected from the wing vein and stored in anticoagulant tubes with heparin. Besides, 10mL blood was collected from the wing vein and stored in the non-anticoagulant tubes, centrifuged (3,500, 4°C, 10 min), divided serum into microcentrifuge tube for reserve.” in lines 140-143 to solve the above problems.

Point 3: In section 2.4 (“Intestinal morphology”) please describe the washing/rinse steps before fixation previously cited in section 2.1 lines 98-100.

Response 3: Thanks for your suggestion. The washing/rinse steps before fixation have been added in lines 105-114, “As intestinal samples, it was collected duodenum, jejunum, and ileum segments taken from midpoint with 2 cm in length. Intestinal tissues were separated, rinsed with sa-line, and placed on formalin were dehydrated, embedded in paraffin, and cut into 5 mm sections. The sections were stained with hematoxylin and eosin by mounting on glass slides and observed under Olympus light microscope (Olympus, Tokyo, Japan) for histological examination. Villus height and crypt depth were measured by Image (Image-Pro Plus 6.1 Media Cybernetics, Rockville, MD), and finally, villus height to crypt depth ratio was calculated.”

Point 4: The authors show an increase in body weight and some lymphoid organs weight, concluding that the ZNPS group presented improvement in feed conversion and immune system response. In addition, ZNPS animals presented an increase in the size of the villa and crypta of all sections of the intestines, which could be correlated to an increase in the weight of the intestines of these birds. Meaning that the animals presented an increase in several organs' weight that could explain the increase in body weight. Was the carcass weight of the broiler chicken measured? If it was it is recommended to be added to the manuscript to back up the conclusion of improvement of feed conversion and growth performance. Without the carcass weight or other cut weight (breast meat, thighs, leg, or wing) it is hard to evaluate if the increase in body weight was because of growth development or just secondary to the increase in weight of internal organs.

Response 4: We gratefully thank you for your precious time in making constructive remarks. During the experiment, we did neglect to detect the carcass weight. Thanks for your advice again, it is very important for us. We will work in the future according to your advice to improve the level of scientific research, and achieve more results. Thank you again for your constructive comments.

Point 5: Line 97-98: “collection of the intestinal sample, and blood and lymphoid organs were harvested” The sentence sounds confusing. Suggested change to: “collection of intestinal and blood samples, and lymphoid organs.”

Response 5: Thanks for your suggestion. According to your advice, “collection of the intestinal sample, and blood and lymphoid organs were harvested” has been changed to “collection of intestinal and blood samples, and lymphoid organs.” in lines 103-104.

Point 6: Line 98-100: “In addition, duodenum, jejunum, and ileum segments 2 cm in length, taken from the midpoint, were separated, rinsed with saline water, and fixed in formalin for histologic study” I suggest this sentence should be split into two sentences to improve clarity: As intestinal samples, it was collected duodenum, jejunum, and ileum segments taken from midpoint with 2 cm in length. Intestinal tissues were separated, rinsed with saline, and fixed in formalin for histologic study.

Response 6: We really appreciate your valuable comments on our manuscript. We have changed the errors that you pointed out in lines 105-114.

Point 7: Table 3 - D42 BW (g) line: Statistical significance letters are 2b, b, and a, the legend does not describe the meaning of “2b”. Is it a typo in the table?

Response 7: We apologize for our mistakes and appreciate your comment. We have superscript all the letters in the table.

Point 8: Section 3.5: In the text and the title of the table is referred to as “Table 4”, but it should be ”Table 6” according to the order in the manuscript.

Response 8: We feel sorry for our mistakes and gratefully appreciate your comments. We have corrected the errors that you pointed out and have carefully checked the full text.

Point 9: Section 3.6: In the text and the title of the table is referred to as “Table 5”, but it should be ”Table 7” according to the order in the manuscript.

Response 9: We apologize for our mistakes and appreciate your comment. We have corrected the errors that you pointed out and have carefully checked the full text.

Point 10: Section 3.7: In the text and the title of the table is referred to as “Table 6”, but it should be ”Table 8” according to the order in the manuscript. In addition, the table should be mice to be between the table title and the legend.

Response 10: We apologize for our mistakes. We have corrected the errors that you pointed out.

Point 11: Line 303: “NC grouop” substitute for “NC group”.

Response 11: Thank you for your comment We have corrected the errors that you pointed out.

Point 12: Line 313: “In broiler fabricus, the spleen…” please, check the spelling. The sentence is confusing, I am not sure what the authors wanted to say.

Response 12: We gratefully appreciate your valuable comment. it is very important, because of your advice, let us have a great harvest. In order to make the article more precise, " ZNPS could significantly improve the morphology of jejunum, ileum, and duodenum" has been changed to " A increase in the intestinal villus height and VH/CD in ZNPS group" in line 363 according to the comments. Besides, we have added

“Studies have shown that Zn increases the activation and proliferation of lymphocytes, primarily T cells and natural killer (NK) cells, and that the immunomodulatory effects of Zn also lead to increased activity of thymocytes, macrophages and heterophil cells, as well as increased antibody production, thereby enhancing the potential for humoral responses [25].” in lines 369-373 for explain how Zn aids in the development of lymphoid organs.

[25] Jarosz L, Marek A, Gradzki Z, Laskowska E, Kwiecien M: Effect of Zinc Sulfate and Zinc Glycine Chelate on Concentrations of Acute Phase Proteins in Chicken Serum and Liver Tissue. Biol Trace Elem Res 2019, 187:258-272.

Point 13: Line 315: “splenic weight is associated with immune cell proliferation.” An increase in spleen size and weight is usually correlated to B cell differentiation and adaptive immune system response. Most immune cell proliferation (including heterophils, macrophages, and B cell progenitors) occurs in the bone marrow. Please, correct it in the text.

Response 13: We feel sorry for our mistakes and gratefully appreciate your comments. “Thymus weight is associated with the generation of developing T cells, and splenic weight is associated with immune cell proliferation.” has been changed to “An increase in spleen size and weight is usually correlated to B cell differentiation and adaptive immune system response.” in lines 374-376.

Point 14: Line 344: “he Firmicutes and Bacteroidetes…” should be substituted for “The Firmicutes and Bacteroidetes”.

Response 14: We apologize for our mistakes and appreciate your comment. We have corrected the errors that you pointed out and have carefully checked the full text.

Point 15: Section 5: The last sentence is missing the final period.

Response 15: We apologize for our mistakes and appreciate your comment. We have corrected the errors that you pointed out and have carefully checked the full text.

Thank you very much for your kind advice.

Round 2

Reviewer 1 Report

Authors perform appreciated revision. Just correct the response for point 2.. add appropriate reference for the technique such as Vollmer, Robin T. "Theory and practice of histological techniques." JAMA 238.25 (1977): 2730-2730.

Author Response

Responses to Reviewer # 1:

Dear Reviewer:

We appreciated all valuable comments from you on our manuscript entitled “Effect of two particle sizes of nano zinc oxide on growth performance, immune function, digestive tract morphology and intestinal microbiota composition in broilers”. Based on your valuable advice, we have amended the relevant part of the manuscript and answered your questions below.

Point 1: Line 138: add reference for the used technique.

Response 1: We gratefully thank you for your precious time in making constructive remarks. We have already added reference for the used technique in lines 151-152, "Fixed tissues were embedded in paraffin and sectioned into 4 μm sections [18], followed by staining with hematoxylin and eosin.” Besides, “Villus height (VH) and Crypt depth (CD) were measured on 8 to 10 well oriented villi and corresponding crypts from each section of all segments of the intestine using a Nikon ECLIPSE 80i light microscope (Nikon Corporation, Tokyo, Japan), and the ratio of villi height/ crypt to depth (VH/CD) was calculated [15].” was added in lines 153-156.

[18] Zhang Q, Zhang K, Wang J, Bai S, Zeng Q, Peng H, Zhang B, Xuan Y, Ding X: Effects of coated sodium butyrate on performance, egg quality, nutrient digestibility, and intestinal health of laying hens. Poult Sci 2022, 101:102020.

[15] Gyawali I, Zeng Y, Zhou J, Li J, Wu T, Shu G, Jiang Q, Zhu C: Effect of novel Lactobacillus paracaesi microcapsule on growth performance, gut health and microbiome community of broiler chickens. Poult Sci 2022, 101:101912.

Thank you very much for your kind advice.

Reviewer 2 Report

The authors have significantly improved the manuscript and revised it in response to the reviewer's inquiries. I have no further comments. 

Author Response

Responses to Reviewer # 2:

Dear Reviewer:

We appreciated all valuable comments from you on our manuscript entitled “Effect of two particle sizes of nano zinc oxide on growth performance, immune function, digestive tract morphology and intestinal microbiota composition in broilers”.

Thank you very much for your kind advice.